# Secalonic Acid-F, a Novel Mycotoxin, Represses the Progression of Hepatocellular Carcinoma via MARCH1 Regulation of the PI3K/AKT/β-catenin Signaling Pathway

**DOI:** 10.3390/molecules24030393

**Published:** 2019-01-22

**Authors:** Lulu Xie, Minjing Li, Desheng Liu, Xia Wang, Peiyuan Wang, Hanhan Dai, Wei Yang, Wei Liu, Xuemei Hu, Mingdong Zhao

**Affiliations:** 1Department of Imaging, Binzhou Medical University, Yantai 264003, China; xielulu0796@163.com (L.X.); wangpeiyuan1640@163.com (P.W.); daihanhan0233@163.com (H.D.); yangwei1172254185@163.com (W.Y.); a414675291@163.com (W.L.); 2Medicine and Pharmacy Research Center, Binzhou Medical University, Yantai 264003, China; liminjing@suda.edu.cn; 3Department of Pharmaceutical Sciences, Binzhou Medical University, Yantai 264003, China; desheng_liu@sina.com; 4Department of Oral Pathology, Binzhou Medical University, Yantai 264003, China; wangxia7512@163.com; 5Department of Immunology, Binzhou Medical University, Yantai 264003, China

**Keywords:** hepatocellular carcinoma, secalonic acid-F, MARCH1, proliferation, PI3K/AKT/β-catenin, magnetic resonance imaging

## Abstract

Liver cancer is a very common and significant health problem. Therefore, powerful molecular targeting agents are urgently needed. Previously, we demonstrated that secalonic acid-F (SAF) suppresses the growth of hepatocellular carcinoma (HCC) cells (HepG2), but the other anticancer biological functions and the underlying mechanism of SAF on HCC are unknown. In this study, we found that SAF, which was isolated from a fungal strain in our lab identified as Aspergillus aculeatus, could inhibit the progression of hepatocellular carcinoma by targeting MARCH1, which regulates the PI3K/AKT/β-catenin and antiapoptotic Mcl-1/Bcl-2 signaling cascades. First, we confirmed that SAF reduced the proliferation and colony formation of HCC cell lines (HepG2 and Hep3B), promoted cell apoptosis, and inhibited the cell cycle in HepG2 and Hep3B cells in a dose-dependent manner. In addition, the migration and invasion of HepG2 and Hep3B cells treated with SAF were significantly suppressed. Western blot analysis showed that the level of MARCH1 was downregulated by pretreatment with SAF through the regulation of the PI3K/AKT/β-catenin signaling pathways. Moreover, knockdown of MARCH1 by small interfering RNAs (siRNAs) targeting MARCH1 also suppressed the proliferation, colony formation, migration, and invasion as well as increased the apoptotic rate of HepG2 and Hep3B cells. These data confirmed that the downregulation of MARCH1 could inhibit the progression of hepatocellular carcinoma and that the mechanism may be via PI3K/AKT/β-catenin inactivation as well as the downregulation of the antiapoptotic Mcl-1/Bcl-2. In vivo, the downregulation of MARCH1 by treatment with SAF markedly inhibited tumor growth, suggesting that SAF partly blocks MARCH1 and further regulates the PI3K/AKT/β-catenin and antiapoptosis Mcl-1/Bcl-2 signaling cascade in the HCC nude mouse model. Additionally, the apparent diffusion coefficient (ADC) values, derived from magnetic resonance imaging (MRI), were increased in tumors after SAF treatment in a mouse model. Taken together, our findings suggest that MARCH1 is a potential molecular target for HCC treatment and that SAF is a promising agent targeting MARCH1 to treat liver cancer patients.

## 1. Introduction

Liver cancer is a major health problem worldwide; there are approximately 841,000 new cases and an estimated 782,000 liver cancer-related deaths annually, making it the sixth most frequently diagnosed cancer and the fourth leading cause of cancer death worldwide in 2018 [1]. There are an array of treatment options for the different stages of HCC, including hepatic surgical resection, transplantation, ablation, chemoembolization, and systemic therapy, according to the Barcelona Clinic Liver Cancer (BCLC) staging system [2,3]. However, most HCC patients are diagnosed at advanced stages and are unable to undergo surgical resection or have poor prognosis after surgical resection [4,5,6]. Therefore, more potent anticancer drugs and treatment options are urgently needed for HCC therapy.

Secalonic acid-F (SAF) is a natural mycotoxin isolated from a natural novel fungal strain. Secalonic acids are natural mycotoxins with a dimeric tetrahydroxanthenone skeleton and display a wide range of anticancer and antimicrobial biological activities, including the inhibition of DNA topoisomerase I and protein kinase C [7]. The secalonic acid family includes dimers of six different monoxanthones (secalonic acids A–F) [8]. Previous studies have reported that SAD could inhibit VEGF-mediated angiogenesis through the Akt/mTOR pathway in breast cancer [9]. Previously, we demonstrated that SAF has anticancer activity, inhibiting the proliferation and promoting the apoptosis of HepG2 cells [10]. However, the underlying mechanisms and other anticancer activities of SAF are unclear.

Recently, the molecular events that drive tumor initiation and progression have attracted increasing attention. Hepatocarcinogenesis is usually related to extensive inflammation and fibrosis responsible for the complex multistep pathogenesis such as disrupted molecular signaling pathways and accumulation of genetic alteration [11,12]. Both the PI3K/AKT/mTOR and the Wnt/β-catenin signaling cascades are known to be important canonical signaling pathways involved in tumor development and progression; approximately 50% of HCC cases display aberrantly activated PI3K/AKT/mTOR and Wnt/β-catenin signaling [13]. The deregulation of these signaling pathways could affect key tumor oncogenes and suppressive genes [14]. Consequently, the discovery of novel molecular targets and agents that inhibit the PI3K/AKT/β-catenin pathway in HCC are pivotal to identifying targeted therapies that could improve HCC patient care. In the present study, we first found that SAF could repress the level of MARCH1. MARCH1, as a membrane-anchored RING-CH1, is a member of the membrane-anchored E3 ubiquitin ligase [15]. Ubiquitination is known as one of the posttranslational modifications that regulates several of the protein degradation and signaling pathways in various inflammatory and cancer diseases, including DNA repair, cell cycle regulation, sodium channel function, immune response and inflammation regulation and cellular stress response regulation, by interacting with such substrates as tumor suppressor proteins, cyclins, and NF-kB [15,16]. Ubiquitination usually involves the close cooperation of three enzymes that activate (E1), conjugate (E2), and ligate (E3) the ubiquitin moiety to the target substrates, and E3 ubiquitin ligases play an essential role in the regulation of ubiquitination [16,17,18]. Very recently, a study demonstrated that MARCH1 was aberrantly overexpressed in ovarian cancer tissues and that knockdown of MARCH1 expression inhibited the progression of ovarian cancer cells via the downregulation of the NF-κB and Wnt/β-catenin pathways [19]. However, the role of MARCH1 in the development and progression of HCC is still not uncovered, and the underlying mechanism is unknown. Therefore, focusing on the role of SAF and MARCH1 in HCC may help us to understand the targeting agents that curtail cancer in the future.

In the present study, we demonstrated that the level of MARCH1 was downregulated in HepG2 and Hep3B cells after treatment with SAF. Importantly, we identified the underlying molecular mechanism by which SAF not only downregulates the expression of MARCH1 but also further inhibits the downstream signaling cascades PI3K/AKT/β-catenin and antiapoptotic Mcl-1/Bcl-2 and activates the cleaved caspase-3 and cleaved caspase-7 signaling pathways to repress the development and progression of HCC with respect to growth, apoptosis, cell cycle, migration, and invasion, both in vitro and in vivo. In addition, we used MARCH1 siRNA to further confirm the anticancer effects of SAF in HepG2 and Hep3B cells and the molecular mechanism of SAF in the regulation of PI3K/AKT/β-catenin signaling pathways. Here, our findings describe the potent antitumor activities of both SAF and MARCH1; moreover, SAF may be a potential therapeutic agent to target MARCH1 in HCC.

## 2. Results

### 2.1. SAF Inhibited HCC Proliferation and Downregulated MARCH1 Expression in HCC Cells

Secalonic acid-F (SAF), as a novel mycotoxin, was isolated from a fungal strain. To explore the therapeutic consequences of SAF in HCC (HepG2 and Hep3B), HepG2 and Hep3B cells were treated with different doses of SAF for 24 or 48 h. The proliferative activities of HepG2 and Hep3B cells were reduced significantly in response to SAF in a dose-dependent manner at 24 h and 48 h, respectively (Figure 1A,B), which exhibited the high toxicity of SAF in HepG2 cells (IC50 = 3.618 μΜ) and Hep3B cells (IC50 = 17.754 μΜ). Surprisingly, we also found that the MARCH1 expression in both HepG2 and Hep3B cells treated with SAF was significantly downregulated in a dose-dependent manner, as measured by western blotting (Figure 1C). These results indicated that SAF could suppress HCC proliferation, and that MARCH1 expression in HCC was associated with the effects of SAF treatment. Moreover, we found that SAF have not downregulated MARCH1 in gene transcription. On the contrary, SAF have slightly upregulated MARCH1 in mRNA level by using qRT-PCR in HepG2 cells (Figure 1D). Interestingly, we found that MARCH1 could be partially stabilized by a proteasome inhibitor (MG132) in HepG2 cells treated with SAF (Figure 1E), indicating that some MARCH1 turnover was mediated by proteasomes. These findings show that MARCH1 may be degraded via proteasome pathway [20]. To examine the role MARCH1 in HCC, two different targeting sequences of siRNA (MARCH1 siRNA-1 and MARCH1 siRNA-2) were used to transiently knock down MARCH1 expression in HCC cells. The blank control (nontransfected) and negative siRNA (transfected negative siRNA) groups were used as the negative controls. As Figure 1F show, MARCH1 silencing inhibited the growth of HepG2 and Hep3B cells compared with the growth of the control groups, and there was no significant difference in the cell growth of the blank control and negative siRNA groups. The knockdown of MARCH1 by siRNA was confirmed by western blot analysis. These findings suggest that MARCH1 knockdown can inhibit HCC proliferation and that MARCH1 plays an important role in the viability of HCC cells. In all, SAF was partially targeting MARCH1 to inhibit the proliferation in HCC cells.

### 2.2. SAF Induced Apoptosis of HCC Cells by Targeting MARCH1

Given some differences in the viability of HepG2 and Hep3B cells in response to the different concentrations of SAF, the concentrations of 1.25, 2.5, and 5 μΜ were selected as appropriate doses to explore the biological function and underlying molecular mechanisms of SAF in both HepG2 and Hep3B cells. We assessed the effect of SAF therapy in HepG2 and Hep3B cells by using a colony formation assay. The number of colonies in the cells treated with 1.25, 2.5, and 5 μΜ SAF was markedly reduced in a dose-dependent manner (Figure 2A). Flow cytometric analysis was also used to analyze the rate of apoptosis in cells that were stained with annexin V and propidium iodine. As shown in Figure 2B, we found that SAF significantly promoted the apoptosis of both HepG2 and Hep3B cells in a dose-dependent manner at 24 h and 48 h, respectively. The number of apoptotic cells increased by 2.8-, 4.2-, and 7.2-fold in HepG2 in response to 1.25, 2.5, and 5 μΜ SAF, respectively, compared to control cells (0 μΜ); similarly, the number of apoptotic cells increased by 3.7-, 8.1-, and 10.9-fold in Hep3B compared to controls. Additionally, we assessed the effect of silencing MARCH1 in HepG2 and Hep3B cells by using a colony formation assay. The same result was clearly verified: the number of colonies was reduced in the cells transfected with MARCH1 siRNA, and no significant difference was found in the number of colonies between the blank control and negative siRNA control. The knockdown of MARCH1 by siRNA in the HepG2 and Hep3B cells were confirmed by western blotting assay (Figure 2C). In addition to the analysis of whether MARCH1 silencing led to cell death, results similar to those from SAF treatment were obtained: the rate of apoptosis was increased in HepG2 and Hep3B cells transfected with MARCH1 siRNA. The number of apoptotic cells increased 1.7-fold in HepG2 cells and 1.8-fold in Hep3B cells in response to MARCH1 siRNA-1, and the number of apoptotic cells increased 2.4-fold in HepG2 cells and 2.6-fold in Hep3B cells in response to MARCH1 siRNA-2 compared to those in negative control cells (negative siRNA), there were no significant differences in the apoptotic rate between the blank control and negative siRNA groups, and the MARCH1 knockdown in HepG2 and Hep3B cells was effective (Figure 2D). These data indicated that SAF downregulated MARCH1 and may enhance apoptosis in HepG2 and Hep3B cells.

### 2.3. SAF suppressed HCC Cell Migration and Invasion

To investigate whether SAF influences HCC cell motility, wound healing assays were performed. The wound healing assays showed that the cell migration of HepG2 cells was suppressed by approximately 27.2% and 41.8% with 2.5 and 5.0 μΜ SAF, respectively, at 6 h and by approximately 69.9% and 79.4% with 2.5 and 5.0 μΜ SAF, respectively, at 24 h; in Hep3B cells, cell migration was decreased by 19.0% and 30.7% at 24 h with 2.5 and 5.0 μΜ SAF, respectively and, by approximately 11.8% and 30.8% at 32 h with 2.5 and 5.0 μΜ SAF, respectively (Figure 3A). 

Similar results were acquired in the HCC cells transfected with MARCH1 siRNA: cell migration was inhibited by approximately 50.1%, 24.6%, and 17.8% at the three time points of 6, 24, and 48 h, respectively, in the HepG2 cells transfected with MARCH1 siRNA-1 compared to negative siRNA controls. Additionally, there were reductions of approximately 32.1%, 30.3%, and 23.6% at 6, 24, and 48 h, respectively, in the HepG2 cells transfected with MARCH1 siRNA-2; in the Hep3B transfected cells, compared to negative siRNA controls, there were reductions of 28.4%, 30.0%, and 20.5% at 6, 24, and 48 h, respectively, with MARCH1 siRNA-1 treatment and reductions of 32.7, 33.8, and 35.3% at 6, 24, and 48 h, respectively, with MARCH1 siRNA-2 treatment the wound healing assays; we did not detect any differences in the cell wound healing ratio between the blank control and negative siRNA groups (Figure 3B). The effect of MARCH1 silencing by siRNA was tested by western blot analysis (Figure 3C).

In addition, SAF significantly inhibited the migration and invasion of both the HepG2 and the Hep3B cells in a dose-dependent manner in the Transwell assays. The migration percentages were reduced to 30.1% and 15.3% of control with 2.5 µM SAF in the HepG2 and Hep3B cells, respectively, and to 14.1% and 4.7% of control with 5.0 µM SAF in the HepG2 and Hep3B cells, respectively (Figure 3D). Additionally, the invasion percentages were reduced from 100% to 67.0% and 62.7% of control with 2.5 µM SAF in the HepG2 and Hep3B cells, respectively, and to 47.2% and 44.8% of control with 5.0 µM SAF in the HepG2 and Hep3B cells, respectively (Figure 3E). We then further evaluated the effect of MARCH1 downregulation on cell migration and invasion. Transwell assays showed impaired migration and invasion of HepG2 and Hep3B cells after MARCH1 knockdown. Compared to those in the blank controls, the migration ratios were reduced to 69.4% and 53.2% in the HepG2 and Hep3B cells, respectively, transfected with siRNA-1 targeting MARCH1 and to 67.1% and 43.1% in the HepG2 and Hep3B cells, respectively, transfected with the siRNA-2 treatment; the MARCH1 knockdown in HepG2 and Hep3B cells was effective (Figure 3F). Whereas the invasion ratios were reduced to 46.1% and 38.3% and 63.5% and 34.8% in the same experimental cells, the downregulation of MARCH1 by siRNA was verified by the western blotting assay (Figure 3G). No differences were exhibited in the Transwell migration and invasion between the blank control and negative siRNA groups. These data revealed that SAF may impair the migration and invasion of HepG2 and Hep3B cells by targeting MARCH1.

### 2.4. MARCH1 SAF Induced HCC Cycle Arrest

Additionally, we tested whether SAF could affect HCC cell cycle arrest. For this experiment, flow cytometric analysis was used to detect the cell cycle distribution. We found that both the cell cycle of the HepG2 and Hep3B cells when exposed to SAF changed, as shown by the cell cycle distribution. SAF significantly induced G2/M arrests in HepG2 cells in a concentration-independent manner (Figure 4 (Aa–c)). 

The result in Hep3B cells was different; here, SAF significantly increased the subG0 DNA fraction (apoptotic population) (Figure 4(Ba–c)). Therefore, these results suggest that SAF could induce cycle arrest, regulating the development and proliferation of HepG2 and Hep3B cells.

### 2.5. PI3K and AKT Mediate the Downregulation of MARCH1 by SAF

In various cancers, PI3K/Akt serve as pivotal regulators of cell metabolic pathways, which are the classic signaling pathways that regulate cell proliferation, apoptosis, migration, and invasion in HCC. To explain how SAF suppressed cell proliferation, migration, and invasion and induced apoptosis in HepG2 and Hep3B cells, we identified the relevant SAF and MARCH1-regulated signaling pathways by using a western blotting assay. As shown in Figure 5A,B, SAF drastically inhibited the expression of MARCH1 and other key proteins, such as PI3K, phosphorylated AKT and β-catenin. The expression of the downstream antiapoptotic targets Mcl-1 and Bcl-2 was significantly decreased, while the expression of the apoptosis-related molecules cleaved caspase-3 and cleaved caspase-7 was markedly and dose-dependently increased in HepG2 and Hep3B treated with 1.25, 2.5, and 5 μΜ SAF.

Additionally, to identify the molecular mechanisms of the MARCH1 knockdown-associated inactivation of PI3K/AKT/β-catenin in HCC, we used western blotting to detect the regulation of relevant downstream molecules. Notably, consistent with the above observations, treatment with MARCH1 siRNA resulted in loss of MARCH1 protein expression. We observed that PI3K, phosphorylated AKT, β-catenin inhibition and reduction in Mcl-1, Bcl-2 protein, whereas the upregulated expression of cleaved caspase-3 and cleaved caspase-7 were exhibited in both HepG2 and Hep3B cells transfected with two sequences of MARCH1 siRNA (Figure 5C,D). Overall, these data demonstrate that MARCH1 is involved in the activation of the PI3K/AKT/β-catenin cascade. The powerful anticancer role of SAF was partially due to mediation of MARCH1 and further regulation of downstream PI3K/AKT/β-catenin signaling pathways in HCC.

### 2.6. SAF Significantly Reduced Tumor Growth in a Xenograft Model of HCC

Next, to confirm this antitumor effect in vivo, a xenograft nude mouse model was employed, and mice were divided into three groups treated with 0, 12.5 and 25 mg/kg SAF with oral intragastric (ig) administration. The body weight and tumor growth were monitored for 5 weeks. Compared with the control group (0 mg/kg/ig), the SAF treatment groups (12.5 mg/kg/ig; 25 mg/kg/ig) showed a more markedly rapid inhibited in tumor volume and tumor weight (Figure 6A–D), while the weight of the animals showed no notable change in the three groups (Figure 6E). These data suggested that SAF suppressed HCC tumor growth in a dose-dependent manner in vivo. These results were further confirmed by MRI, H-E staining, and western blot analyses for the anticancer effects of SAF.

MRI was used to research the tumor responses to SAF therapy in our study. The apparent diffusion coefficient (ADC), as a quantitative parameter of DW-MRI, is usually used as an early prognosis marker in various cancers [21]. Therefore, the representative T2-weighted MRIs and ADC maps of DW-MRI in the three mouse groups were clearly examined. An increase in the hyperintensity of the necrotic area and a decrease in the volume of the tumors treated with SAF are shown with the T2WI (Figure 6F). Interestingly, we found that the apparent diffusion coefficient (ADC) values acquired from the ADC maps in the SAF-treated tumors were significantly higher than those in the control tumors (Figure 6G). A significant decrease in the tumor volume measured from the T2-weighted images was observed (Figure 6H). The negative correlation between the ADC value and tumor volume was validated (Figure 6I). Additionally, compared with that in the control group, H–E staining in the treatment groups showed much more tumor necrosis area and looser cell spacing (Figure 6J). Furthermore, western blotting was performed to detect the expression of the PI3K-related pathway proteins from the tumors treated with different doses of SAF. The expression levels of key proteins in the Akt signaling cascade, such as PI3K, phosphorylated AKT and β-catenin, as well as the levels of the downstream proteins Bcl-2 and Mcl-1, were significantly inhibited in SAF-treated tumors, while the levels of pro-apoptosis-related cleaved caspase-3 and cleaved caspase-7 were elevated (Figure 6K). Thus, these results further suggest that SAF suppressed tumor growth and promoted tumor apoptosis, which was partially due to the reduction in MARCH1 leading to the inhibition of PI3K/AKT/β-catenin signaling pathway in HCC.

## 3. Discussion

The initiation and progression of hepatocellular carcinoma is a complex multistep process that involves the accumulation of several genomic alterations and the deregulation of several signaling pathways. This leads to the aberrant expression of suppressor genes and oncogenes [2,11,22]. Molecular studies have identified that the majority of hepatocellular carcinomas show aberrantly activated PI3K/AKT/mTOR and Wnt/β-catenin signaling pathways [13,23,24]. Consequently, the discovery of novel targeted molecular therapeutic agents and key pathway inhibitors is pivotal to the identification of therapeutic protocols to improve HCC patient treatment and care. To date, the antitumor effects and mechanism of SAF have remained unexplored. In the present study, we explored its targeted molecule and its anticancer effects on cell proliferation, apoptosis, cell migration, cell invasion, and the cell cycle as well as on the PI3K/AKT/β-catenin signaling cascade in HepG2 and Hep3B cells. Furthermore, we first identified that the cells treated with SAF could suppress the level of MARCH1 expression, which may be an important factor for HCC progression. 

Membrane-associated RING-CH1 (MARCH1) is a member of the MARCH family, which is an E3 ubiquitin ligase [25]. Ubiquitination is known as a key regulator of various signaling pathways and protein degradation [16,17]. As reported, MARCH1 can mediate the ubiquitination of MHC-II, CD86, Fas, and CD98, leading to immune suppression [15,26,27]. Until now, the role of MARCH1 in the development and progression of HCC was unexplored; futhermore, the relationship between MARCH1 and SAF was unknown. However, in the current study, we found that MARCH1 downregulation has a powerful anticancer effect in HepG2 and Hep3B cells. In addition, MARCH1 was significantly downregulated in response to SAF in a dose-dependent manner in HCC cells. The inhibition of MARCH1 was also detected in SAF-treated tumors in vivo. Moreover, downregulated MARCH1 in response to SAF was not via gene transcription, but may be degraded via proteasome pathway. Our data suggest that MARCH1 may be a novel molecular target for SAF. Then, the key question of the underlying molecular mechanisms of SAF treatment arose and needed to be further explored.

The phosphoinositide 3-kinase (PI3K)/AKT signaling pathway is considered an important regulator in HCC and affects various signaling and biological processes. The activation of PI3K/AKT signaling results in HCC cell proliferation, migration, invasion and cycle arrest but inhibits apoptosis [28,29]. Here, both in vitro and in vivo western blotting assays demonstrated that the protein expression levels of PI3K and P-AKT were suppressed in HCC with SAF therapy. Furthermore, functional assays confirmed that SAF suppresses cell proliferation, migration, and invasion and induces cell apoptosis and cycle arrest of hepatocellular cancer in vitro. As expected, the role of SAF in inhibiting HCC cell growth was further identified in a mouse model. MARCH1 knockdown was also found to reduce the protein expression of PI3K and P-AKT and further inhibited cell proliferation, migration, invasion, and induced apoptosis of HCC. Recently, genomics studies have reported that β-catenin is a key oncogenic driver that plays a major role in tumorigenesis and the development of HCC [13,30]. It is known that the activation of AKT mediates the phosphorylation of GSK3β, resulting in the nuclear translocation of β-catenin from the cytoplasm and increased transcription, leading to the regulation of downstream cyclin D1, c-Myc, and CD44 protein expression, which ultimately inhibits HCC tumor growth [31]. Conversely, enhanced Wnt/β-catenin signaling supports the growth of hepatocellular carcinoma [32]. The inhibition of oncogenic β-catenin signaling repressed tumor growth in prostate cancer [33], as well as in lung adenocarcinoma [34] and breast cancer [35]. In our study, we analyzed the effect of SAF on β-catenin by detecting the protein expression level of β-catenin. Our results suggested SAF treatment induced not only the inhibition of cell cycle progression, migration, and invasion but also the reduction in β-catenin in HCC cells. In immune cells, the functions of MARCH8 are closely related to MARCH1 [24]. Sh-MARCH8 reduced cell viability and induced cell apoptosis through blocking the activation of the PI3K/β-catenin signaling axis in gastric cancer [36]. However, MARCH1 downregulation also inhibited the activation of β-catenin in HCC cells. Our result was consistent with a previous study, which showed that MARCH1 knockdown induced the inhibition of β-catenin/NF-κB and resulted in the suppression of ovarian cancer [19]. These results demonstrated the powerful anticancer effect of SAF partially through targeting MARCH1, which inhibited PI3K/AKT/β-catenin signaling. The apoptotic mechanism of HCC is a complex process and involves various pathways in which any imbalance may induce malignant transformation and tumor metastasis and may even result in resistance to anticancer drugs. The Bcl-2 family proteins play a major role in the progression of various malignancies. These proteins are divided into two categories: antiapoptotic (Bcl-2, Bcl-xL, Mcl-1, Bcl-w and A1) and proapoptotic (NOXA, PUMA, Bim and Bid). Among the antiapoptotic members, Mcl-1 and Bcl-2 are antiapoptotic members of the Bcl-2 family and are key survival factors in various neoplasms [37,38]. Mcl-1 plays a critical role in chemoresistance and tumorigenesis, particularly in solid cancers, such as non-small-cell lung cancer [39], oral squamous cell carcinomas [40], Non-Hodgkin's B-cell lymphomas [41], and hepatocellular carcinoma [42]. Therefore, we further explored the mechanism that might be responsible for SAF-induced apoptosis. The antiapoptotic Mcl-1 and Bcl-2, which involve mitochondrial apoptosis, both showed a drastic reduction in HepG2, Hep3B cells and xenografts in response to SAF treatment. Subsequently, SAF treatment also significantly induced caspase-3 and caspase-7 activation, which are involved in the mitochondrial-dependent apoptosis pathway. In addition, further study showed that the downregulation of MARCH1 suppressed the expression of the antiapoptotic proteins Mcl-1 and Bcl-2 in vitro in HepG2 and Hep3B cells but enhanced the activation of cleaved caspase-3 and cleaved caspase-7. Consistent with recent studies, the inhibition of the PI3K/AKT pathway significantly induces antiapoptosis and promotes survival through Mcl-1 and Bcl-2 degradation in AML [43], human myeloid leukemia cells [44], and HCC [45]. These data suggest that MARCH1 plays a major role in the development and progression of HCC. SAF partially targeted MARCH1 to inhibit HCC growth via PI3K/AKT/β-catenin inactivation and Mcl-1/Bcl-2 downregulation (Figure 6L).

Moreover, in vivo, we found that SAF treatment markedly reduced tumor growth in an HCC xenograft mouse model. Here, we demonstrated that SAF affected the biomarkers of HCC tumors. The ADC value, derived from diffusion-weighted imaging (DWI), is a sensitive biomarker for predicting early treatment responses in various malignant tumors that was used to assess tumor cellularity, tumor necrosis, cell density and apoptosis [21,46]. Previous studies reported that ADC values were increased after cisplatin treatment in an ovarian cancer model [47], after sorafenib treatment in breast cancer xenografts [48], and after irinotecan treatment in colon carcinoma xenografts [49]. Hence, we used DW-MRI to monitor the ADC values in HCC xenografts in response to SAF therapy. Interestingly, our findings are consistent with previous studies. The ADC values were increased after SAF treatment in tumors. The increase in ADC values was attributed to tumor cell death and apoptosis. These results were supported by the H–E and western blotting assays. Increased tumor necrosis and less compact cell spaces were observed in SAF-treated tumors with H–E staining, and higher levels of the proapoptotic proteins cleaved caspase-3 and cleaved caspase-7 were detected in SAF-treated tumors by western blotting. Therefore, our findings further suggest that SAF plays a vital role in the antitumor effect of HCC.

## 4. Materials and Methods 

### 4.1. Cell Culture

The human HCC cell lines (HepG2, Hep3B) were purchased from the Cell Research Institute of the Chinese Academy of Sciences (Shanghai, China). The cells were cultured in Dulbecco’s modified Eagle medium (DMEM) with high glucose (HyClone, Logan, UT, USA) supplemented with 10% fetal bovine serum (FBS; Gibco, Waltham, MA, USA), penicillin (100 units/mL, Solarbio, Beijing, China), and streptomycin (100 μg/mL, Solarbio) at 37 °C in a humidified atmosphere with 5% CO_2_. 

### 4.2. Reagents and Antibodies

SAF was isolated from a fungal strain in a lab from the Department of Pharmaceutical Sciences of Binzhou Medical University. SAF was dissolved in dimethyl sulfoxide (DMSO) and was stored at −20 °C as a stock solution. MARCH1 (bs-9335, Bioss, Beijing, China), phospho-AKT Ser473 (Sikh Association of Baltimore, Randallstown, MD, USA), total AKT (21054, SAB), GAPDH (10494-1-AP, Proteintech, Wuhan, China), PI3K (20584-1-AP, Proteintech, Tokyo, Japan), β-catenin (51067-2-AP, Proteintech), Mcl-1 (16225-1-AP, Proteintech), Bcl-2 (12789-1-AP, Proteintech), cleaved caspase-3 (9661, Commonwealth Soap & Toiletries, Fall River, MA, USA), cleaved caspase-7 (8438 CST), MG 132 (HY-13259, MedChem Express, Monmouth Junction, NJ, USA) and secondary antibodies (peroxidase-conjugated goat anti-rabbit IgG; ZB-2301, ZSGB-BIO, Beijing, China) were obtained commercially.

### 4.3. siRNA Transfection

The siRNAs against MARCH1 were produced by Gene Pharma (Shanghai, China). MARCH1 siRNA-1 and MARCH1 siRNA-2 are two different siRNA sequences targeted to different sites in MARCH1. The negative siRNA was used as a control for all siRNA experiments. The sequences for all the siRNAs are as follows: for MARCH1 siRNA-1, the sense sequence was 5′-CAGGAGGUCUU GUCUUCAUTT-3′, and the antisense sequence was 5′-AUGAAGACAAGACCUCCUGTT-3′; for MARCH1 siRNA-2, the sense sequence was 5′-GGUAGUGCCUGUACCACAATT-3′, and the antisense sequence was 5′-UUGUGGUACAGGCACUACCTT-3′; and regarding the negative siRNA, the sense sequence was 5′-UUCUCCGAACGUGUCACGUTT-3′, and the antisense sequence was 5′-ACGUGACACGUUCGGAGAATT-3′. The cells were transfected with 60 nM siRNA-MARCH1 in 2 mL serum-free medium without FBS and 6 µL Lipofectamine 2000 (Thermo Fisher, Waltham, MA, USA) according to the manufacturer’s instructions. Finally, western blotting was used to test the knockdown efficiency over 48 h.

### 4.4. Western Blot Analysis

Western blot analysis was performed as previously reported [50]. Protein extracts of the tissues and cells were obtained using ice-cold RIPA lysis buffer (50 mM Tris pH 7.4, 150 mM NaCl, 1% Triton X-100, 1% sodium deoxycholate, 0.1% SDS, protease inhibitor [Beyotime, Shanghai, China]) containing phenylmethanesulfonyl fluoride (PMSF, Beyotime). Protein extracts were boiled in 5× SDS-PAGE sample loading buffer (Beyotime) for 10 min, and protein concentration was determined using a BCA protein assay kit (Beyotime). Up to 15 µg of protein were separated in SDS-PAGE gels (Beyotime) by electrophoresis, and proteins were transferred to polyvinylidene difluoride (PVDF) membranes (Solarbio, Beijing, China). After membrane blocking in 5% nonfat milk for 2–3 h, the membranes were incubated with the primary antibodies overnight at 4 °C and then washed three times for 30 min in TBST (Tris-buffered saline and Tween-20) before incubation with the secondary antibodies (ZB-2301; Beijing Zhongshan Golden Bridge Technology Co., Ltd., Beijing, China) conjugated with horseradish peroxidase for 2 h at room temperature. The membranes were washed three times with TBST and further quantified using chemiluminescence with a super enhancer ECL kit (Novland, Shanghai, China). The quantification of western blot bands was performed using ImageJ. All experiments were performed at least three time.

### 4.5. Cell Proliferation Assay

Cell proliferation assays were performed using a Cell Counting Kit-8 (CCK-8; Biosharp, Beijing, China) according to the manufacturer’s instructions and as described previously [51]. Briefly, 5 × 10^3^ cells per 96-well plate were cultured overnight and incubated with different concentrations of SAF for the indicated time points. After incubation with the CCK-8 reagent for 1 h at 37 °C, the absorbance of samples at 450 nm was measured by a microplate reader (SpectraMax M2, Shanghai, China). All experiments were carried out in triplicates.

### 4.6. Colony Formation Assay

Colony formation was performed as described previously [9,52]. Briefly, 5 × 10^3^ HCC cells were seeded in 6-well plates and grown for over 12 days. The colonies were stained with 0.1% crystal violet solution (Solarbio) and imaged with a photomicroscope. Colonies containing over 50 cells were counted. All experiments were carried out in triplicates.

### 4.7. Flow Cytometric Analysis

For apoptosis analysis, the level of apoptosis of HepG2 and Hep3B cells induced by SAF reagents and MARCH1 siRNA was analyzed using an FITC Annexin V and propidium iodide (PI) apoptosis detection kit (KeyGEN Biotech, Nangjing, China) according to the manufacturer’s instructions and as described previously [53]. The fluorescence intensity of over 1 × 10^4^ cells was analyzed using a flow cytometer (FACSCanto II, Becton Dickinson, Franklin Lakes, NJ, USA). 

Cell cycle analysis was carried out as described previously [54]. HepG2 and Hep3B cells were treated for 24 h and 48 h, respectively, with 1.25, 2.5, and 5.0 µM SAF. To examine the distribution of HepG2 and Hep3B cells in the sub-G1, G0/G1, S, and G2/M phases of the cell cycle, the SAF-treated cells were stained with propidium iodide (PI; Sikh Association of Baltimore) in the presence of RNase A (Sikh Association of Baltimore). Fluorescence intensity was determined using a flow cytometer (BD FACSCanto II). All experiments were carried out in triplicates.

### 4.8. Wound Healing and Transwell Assays

The wound healing process was performed as described previously [55]. Briefly, the HepG2 and Hep3B cells reached 90% confluence, and the cells were scraped with a 200-µl pipette tip to create wounds and were supplemented with fresh medium with 1% FBS to inhibit proliferation. The cells were photographed immediately after scratching and at several indicated times by using a photomicroscope at 100× magnification. The wound areas were assessed by using Image-Pro plus 6.0. The rate of wound healing migration was calculated based on the change in the wound sizes.

Transwell migration and invasion assays were performed by using 6.5 mm inserts and 8.0 µm pore polycarbonate membranes (Corning Inc., Corning, NY, USA,) as described previously [55,56]. Briefly, for the migration assay, 5 × 10^4^ cells were seeded in the upper chamber and cultured in serum-free medium, and the lower chamber was supplemented with 20% FBS medium. For invasion assays, the pore polycarbonate membrane was coated with Matrigel (Corning Inc.). After incubation for 24 h at 37 °C, the upper chamber was washed, dried, fixed with 4% paraformaldehyde, and stained with 0.1% crystal violet, and the cells were counted with a photomicroscope at 400× magnification. All experiments were carried out in triplicates.

### 4.9. Xenograft Models and Treatment

Animal experiments were performed in accordance with the guidelines established by the University of Binzhou Medical Institutional Animal Care and Use Committee. Animal protocols were approved by the Animal Experimental Ethics Committee of the Binzhou Medical University (SYXK 2013 0020). The 4-week-old immunodeficient female BALB/C nude mice were obtained from Vital River Laboratories (Beijing, China). All the mice were housed in the Specific Pathogen Free Animal Laboratory of Binzhou Medical University, which has a 12 h alternating light-dark cycle and temperature-controlled room, and were allowed approximately 1 week of acclimatization to their new surroundings. Previous studies have demonstrated that HCC HepG2 cells could form subcutaneous tumors in the nude mouse model [57,58]. Briefly, as previously described [57,58,59], 1 × 10^7^ HepG2 cells were subcutaneously injected into the left back near the hind leg of the nude mice. When the tumor volumes reached approximately 200 mm^3^, the mice with equivalently sized tumors were randomly divided into 3 groups for different SAF treatments. SAF was dissolved in distilled water containing 6.0% ethyl alcohol absolute and 1.0% Tween 80 and was intragastrically (Ig) administered at doses of 0 mg/kg, 12.5 mg/kg, and 25 mg/kg. The treatment was administered once every two days. The tumor volumes were measured twice per week and calculated as A × B^2^ × 0.5, where A and B represent the length and width of the tumor, respectively. The mice were also weighed every two days. After 5 weeks of therapy, all mice were subjected to magnetic resonance imaging (MRI). Afterward, the mice were sacrificed, the tumors were excised and weighed, and the tumor tissues were harvested for immunoblot analysis and H-E staining.

### 4.10. Magnetic Resonance Imaging (MRI)

Small animal MRI was performed by using a high field 7.0 Tesla MRI system (Bruker BioSpec 70/20USR, Karlsruhe, Germany). During the scan, the mice were placed in an animal bed equipped with circulating warm water to constantly regulate body temperature and were anesthetized with 1–2% inhaled isoflurane (Ruiward Life Technology Co., Ltd., Shenzhen, China). T2-weighted imaging (T2WI) and diffusion-weighted imaging (DWI) of the mice was performed by using a nonmagnetic stereotactic and cylindrical surface wrist coil with a 5.0 cm internal diameter positioned directly over the xenograft area according to the following protocol. The fast spin echo (FSE) T2-weighted images with fat saturation were performed by using the following parameters: TR, 1986.57 ms; TE, 34.37 ms; echo spacing, 11.457 ms; orientation, axial; section thickness, 1 mm; 15 slices; matrix, 512 × 512; and field of view, 40 × 40. An axial respiration-induced single-shot echo diffusion-weighted sequence with fat saturation was performed by using the following parameters: b values, 650 s/mm^2^; TR, 2500.00 ms; TE, 33.00 ms; slice thickness, 1 mm; 15 slices; matrix, 128 × 128; and field of view, 40 × 40. The total acquisition time was approximately 30 min.

### 4.11. Quantitative Real-Time PCR (qRT-PCR)

The HepG2 cells were treated with SAF (5.0μΜ) for 24 h. Total RNA was extracted with RNAiso Plus reagent (Takara, Kusatsu, Japan). All cDNA were reversely transcribed with a PrimeScript^TM^ RT reagent Kit (Takara). Quantitative real-time PCR was conducted using LightCycler system according to the manufacturer's instruction. The MARCH1 primers were produced by Bao Bioengineering Co., Ltd (Takara, Dalian, China). The MARCH1 primers sequences were: 5′-CTGCTGTGAGCTCTGCAAGTATGA-3′ (Forward) and 5′-TACGTGGAATGTGACAG AGCAGAA-3′ (Reverse). The GAPDH primers sequences were: 5′-GCACCGTCAAGGCTGAGAAC-3′ (Forward) and 5′-TGGTGAAGACGCCAGTGGA-3′ (Reverse). The GAPDH was used for normalization. All the primers were produced by Takara. All experiments were carried out at least three time.

### 4.12. Hematoxylin and Eosin (H–E)

The nude mice were sacrificed after MRI, and the tumor tissues were fixed in 4% paraformaldehyde, dehydrated, and embedded into paraffin wax blocks. The embedded tissues were cut into 5-µm-thick sections, placed on adhesion microscope slides (Citoglas, Jiangsu, China) and stained with hematoxylin staining solution and an eosin staining kit (Novland, Shanghai, China).

### 4.13. Statistical Analysis

The statistical analyses were conducted using SPSS 17.0 (SPSS, Chicago, IL, USA) and PRISM 5.0 (5.0 software, GraphPad Prism, San Diego, CA, USA). Differences in means were determined and analyzed by Student’s t-test (two-tailed) and one-way ANOVA (for over 2 groups). All data are presented as the means ± SD. * *p* < 0.05 and ** *p* < 0.01 were considered statistically significant.

## 5. Conclusions

In conclusion, our findings demonstrate that SAF effectively suppresses the development and progression of HCC in vitro and in vivo. Furthermore, our study also suggests that MARCH1 could inhibit the development and progression of HCC. These effects are associated with the inactivation of PI3K/AKT/β-catenin signaling that is mediated by targeting MARCH1, which further induces the loss of Mcl-1 and Bcl-2, leading to accumulation of caspase family member proteins to suppress HCC proliferation, migration, and invasion and to induce cell cycle arrest and apoptosis. These results indicate that MARCH1 plays an important promotion role in HCC progression and may serve as a novel therapeutic target for HCC. This study not only provides a better understanding of how SAF exerts its anticancer activity but also contributes to the role of novel drug discovery and development in the treatment of HCC.

## Figures and Tables

**Figure 1 molecules-24-00393-f001:**
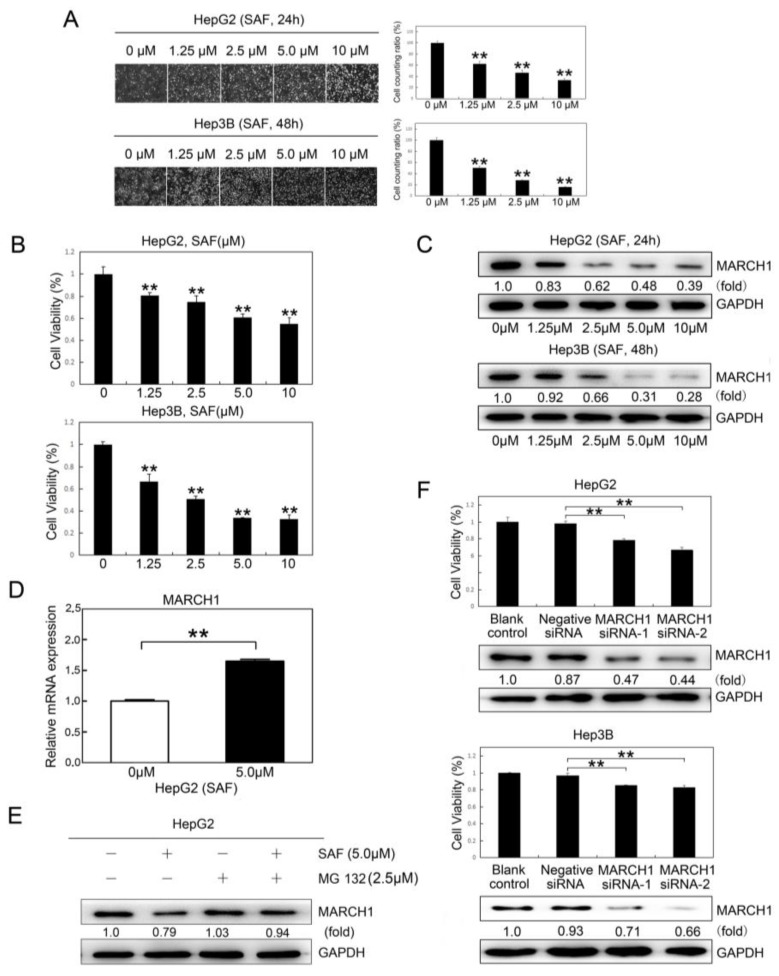
Effect of SAF on HCC cell proliferation. (**A**) Representative images of human HepG2 and Hep3B HCC cells treated with 0, 1.25, 2.5, 5.0, and 10 μM SAF for 24 h and 48 h, respectively. (**B**) Cell viability assay of SAF-treated HepG2 and Hep3B cells for 24 h and 48 h, respectively (0, 1.25, 2.5, 5.0, and 10 μM). The cell growth ratio was determined relative to the untreated control (0 μΜ). (**C**) The expression of the MARCH1 response to SAF in HepG2 and Hep3B cells for 24 h and 48 h was determined by western blotting. (**D**) The MARCH1 mRNA expressions in HepG2 cells with 0, 5,0 μΜ SAF were measured directly by qRT-PCR, GAPDH as an internal control. (**E**) The HepG2 cells were pretreated for 5 h MG 132 which is a proteasome inhibitor. Then, the MARCH1 protein expressions in HepG2 cells treated with 0 μΜ, 5,0 μΜ SAF, and 2.5 μΜ MG 132 were measured by immunoblotting, GAPDH was used as an internal control. (**F**) CCK-8 assays of transfected and nontransfected MARCH1 siRNA of HepG2 and Hep3B cells for 48 h, negative siRNA as control. The knockdown of MARCH1 protein was confirmed by western blot. All data in this figure are presented as means ± SD of three independent experiments. ** *p <* 0.01.

**Figure 2 molecules-24-00393-f002:**
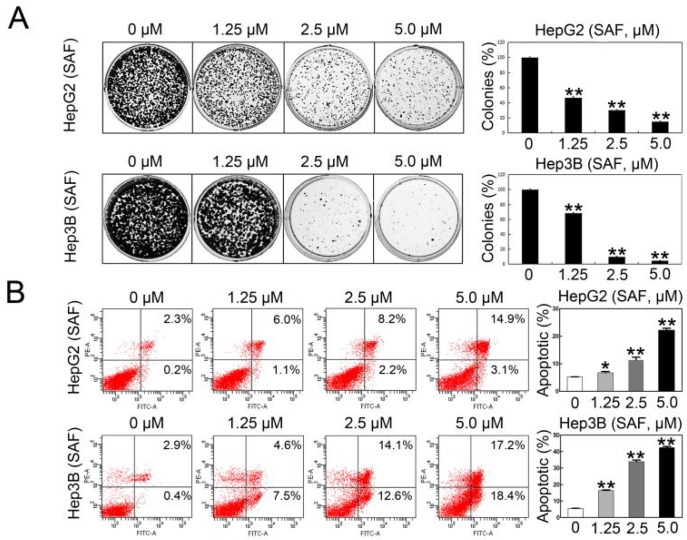
Effect of SAF on HCC cell apoptosis. (**A**) Colonies were stained with crystal violet solution as described in the Materials and Methods. Colony formation analysis of HepG2 and Hep3B cells treated with 0, 1.25, 2.5, and 5.0 μM SAF for 24 h and 48 h, 0 μM as control. (**B**) Flow cytometric analysis of apoptosis in HepG2 and Hep3B cells treated with 0, 1.25, 2.5, and 5.0 μM SAF for 24 h and 48 h. The quantification of apoptotic cells was determined, 0 μM as control. (**C**) Colony formation analysis of HepG2 and Hep3B cells treated with two sets of MARCH1 siRNA, negative siRNA, and non transfected for 48 h, negative siRNA as control. Western blotting was used to confirm the MARCH1 siRNA knockdown in HepG2 and Hep3B cells. (**D**) Flow cytometry showed the apoptosis rate of HepG2 and Hep3B cells treated with MARCH1 siRNA, negative siRNA, and nontransfected for 48 h, negative siRNA as control. Western blotting was used to confirm the MARCH1 silencing efficiency in HepG2 and Hep3B cells. All data in this figure are presented as means ± SD. ** *p <* 0.01, * *p <* 0.05. These data represent three independent experiments.

**Figure 3 molecules-24-00393-f003:**
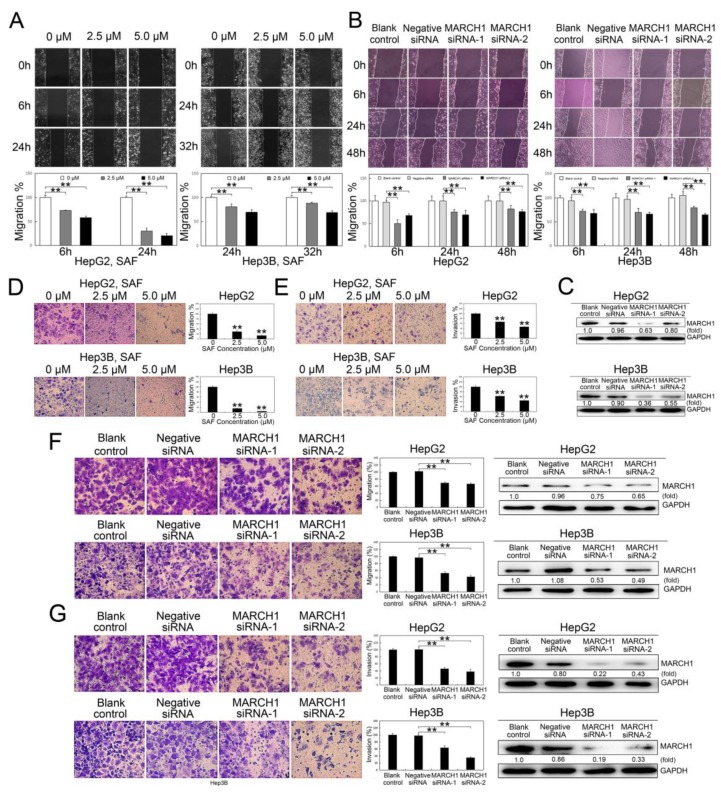
Effect of SAF on HCC cell migration and invasion. (**A**) Wound healing assay for HepG2 and Hep3B cells after SAF treatment, 0 μM as control. (**B**) Wound healing assay in HepG2 and Hep3B cells with MARCH1 interference, negative siRNA as control. (**C**) Knock down of MARCH1 protein with siRNAs in HepG2 and Hep3B cells. (**D**) Transwell migration assay for HepG2 and Hep3B cells after SAF therapy, 0 μM as control. (**E**) Transwell invasion assay for HepG2 and Hep3B cells in response to SAF, 0 μM as control. (**F**) In vitro Transwell migration of HepG2 and Hep3B cells transfected with MARCH1 siRNA, negative siRNA as control. Western blotting was used to assess the silencing efficiency of siRNA against MARCH1 in HepG2 and Hep3B cells. (**G**) In vitro Transwell invasion assay in HepG2 and Hep3B cells after MARCH1 interference, negative siRNA as control. The efficiency of MARCH1 silencing was assessed by western blot. All data in this figure are presented as means ± SD. ** *p <* 0.01. These data represent three independent experiments.

**Figure 4 molecules-24-00393-f004:**
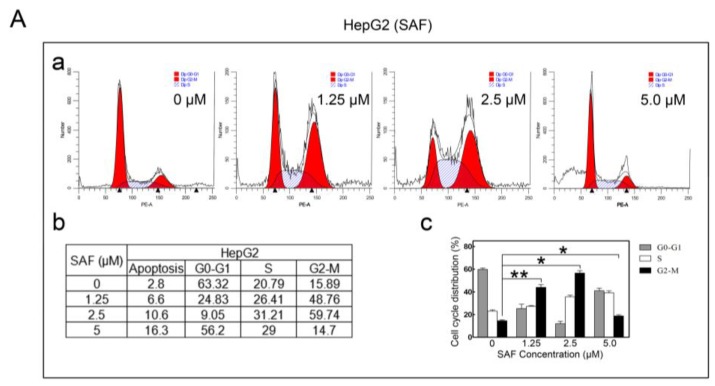
Effect of SAF on the cell cycle. (**A**) Cells were stained with propidium iodide as described in the Materials and Methods and detected by flow cytometry. Cell cycle distribution of SAF-treated or untreated HepG2 cells for 24 h. (**B**) Cell cycle distribution in SAF-treated or untreated Hep3B cells for 48 h. All experiments were carried out in triplicate. The data in this figure are presented as the means ± SD. 0 μM as control, ** *p <* 0.01, * *p <* 0.05.

**Figure 5 molecules-24-00393-f005:**
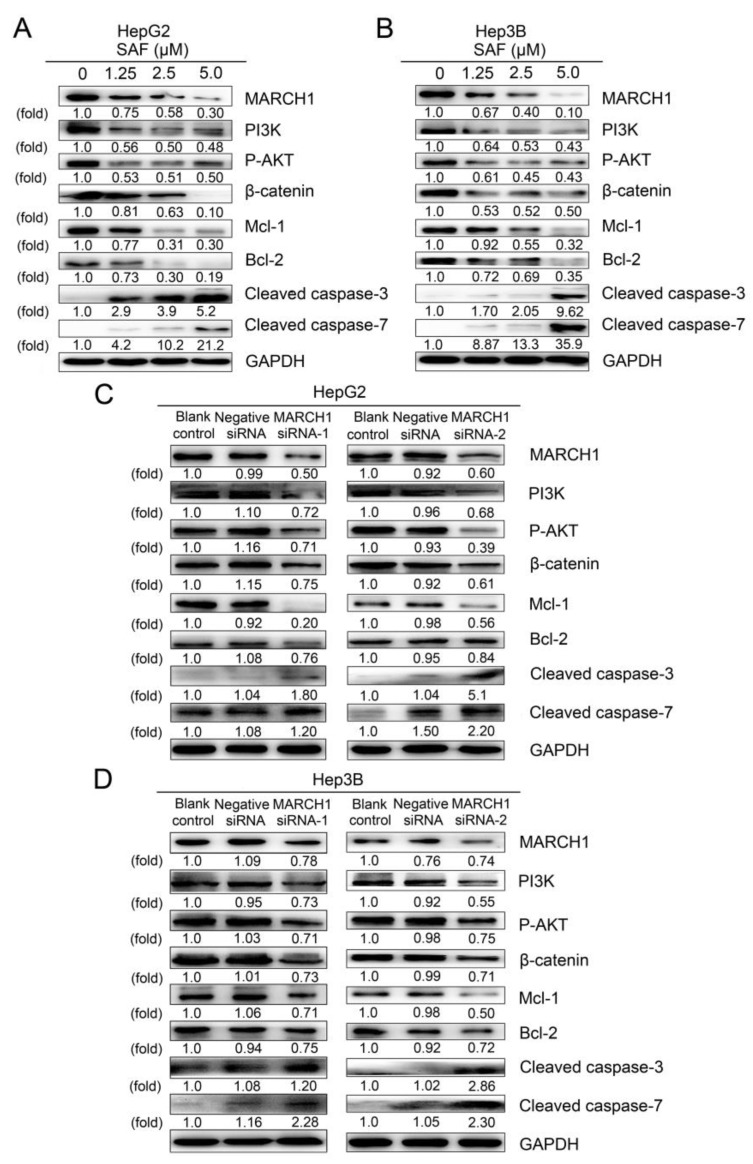
PI3K/AKT/β-catenin involved SAF-mediated suppression of MARCH1 expression in HCC. (**A**,**B**) Western blotting showed the expression of MARCH1, PI3K, P-AKT, β-catenin, Mcl-1, Bcl-2, cleaved caspase-3, and cleaved caspase-7 in HepG2 after 24 h and in Hep3B cells after 48 h of SAF treatment (0, 1.25, 2.5, and 5.0 μM). (**C**,**D**) Protein expression of MARCH1, PI3K, P-AKT, β-catenin, Mcl-1, Bcl-2, cleaved caspase-3, and cleaved caspase-7 in HepG2 and Hep3B cells transfected with MARCH1 siRNA (siRNA-1 and siRNA-2), as measured by western blotting. All data were carried out at least three times. All data in this figure are presented as means ± SD.

**Figure 6 molecules-24-00393-f006:**
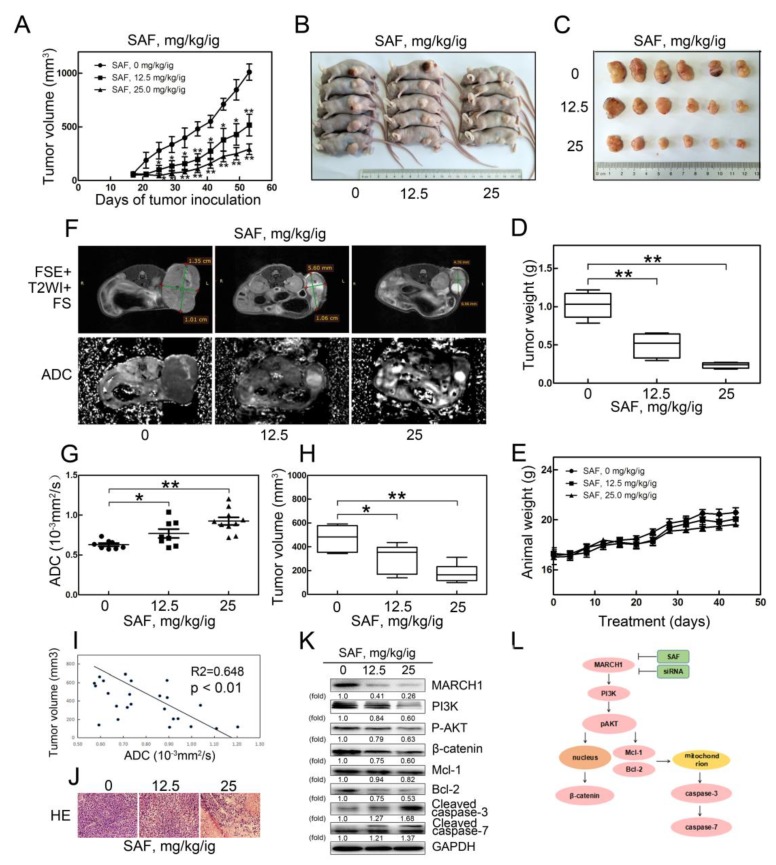
SAF significantly reduces HCC xenograft tumor growth via the MARCH1-mediated inhibition of the downstream PI3K/AKT/β-catenin pathway in vivo. (**A**) Tumor growth curves of the three SAF therapy groups (0 mg/kg/ig, 12.5 mg/kg/ig, 25 mg/kg/ig). (**B**–**C**) Representative images of mice and tumors in the three SAF treatment groups. (**D**) Tumor weight in the untreated and treated mouse groups. (**E**) Body weight changes of mice in the untreated and treated groups. (**F**) T2-weighted MR images and ADC maps of the SAF-treated mice. (**G**) Average ADC values of tumors from untreated and treated mice. (**H**) Average volume of tumors from untreated and treated group mice measured on T2WI. (**I**) Negative correlation between ADC value and tumor volume. (**J**) Representative images of H-E staining in HCC xenograft tumor sections, magnification 40x. (**K**) Protein expression of MARCH1, PI3K, P-AKT, β-catenin, MCL-1, BCL-2, cleaved caspase-3 and cleaved caspase-7 in xenografts of SAF untreated and treated mice. (**L**) A schematic diagram illustrating the underlying anticancer mechanism of SAF on hepatocellular carcinoma, as measured by western blotting. All data in this figure are presented as means ± SD. ** *p <* 0.01, * *p <* 0.05.

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
