# Peer review of "Secalonic Acid-F, a Novel Mycotoxin, Represses the Progression of Hepatocellular Carcinoma via MARCH1 Regulation of the PI3K/AKT/β-catenin Signaling Pathway"

_molecules, 2019, doi:10.3390/molecules24030393_

Round 1

Reviewer 1 Report

In this manuscript, the Authors found that secalonic acid-F (SAF) isolated from a fungal strain identified as Aspergillus aculeatus, may inhibit the progression of hepatocellular carcinoma by targeting MARCH1, along with the PI3K/AKT/β-catenin and the Mcl-1/Bcl-2 signaling

pathways. In this regard, SAF was shown to be able to reduce the proliferation, the migration and invasion, the colony formation of HepG2 and Hep3B cell lines as well as to induce cell apoptosis.

Moreover, the Authors found that MARCH1 is downregulated upon exposure to SAF through the

PI3K/AKT/β-catenin transduction signaling. In addition, the knockdown of MARCH1 by siRNAs suppressed the proliferation, colony formation, migration and invasion and triggered via Mcl-1/Bcl-2 the apoptotis of HepG2 and Hep3B. In vivo experiments confirmed the findings observed in vitro.

On the basis of these results, the Authors concluded that SAF may be considered as a promising  agent targeting MARCH1 toward liver cancer treatment.

The issue is interesting as it deals with the potential use of SAF in patients with hepatocellular carcinoma, however several concerns remain to be addressed, as detailed below.

Figure 1A-B do not mirror the graphical representations shown in figure 1B-C, hence figure 1A-B should be shown in an adequate manner to better evaluate these results.

In Figure 1C-D the cell viability upon SAF treatment is reduced more than using the siRNA1 and the siRNA2 as shown in figure 1E-F (particularly in Hep3B cells), making the results unclear.

In Figure 5 the Authors deal with the potential transduction mechanisms involved in the action of SAF, arguing that  PI3K and AKT mediate the downregulation of MARCH1 by SAF. The Authors only showed the regulation of these pathways by SAF but not their connection. A similar comment regards the downregulation of the antiapoptotic Mcl-1/Bcl-2.

Author Response

Comments from Reviewer 1:

1. Figure 1A-B do not mirror the graphical representations shown in figure 1B-C, hence figure 1A-B should be shown in an adequate manner to better evaluate these results.

Response: Thank you for your professional suggestion and we have adjusted and improved Figure 1 A-D to better evaluate these results in Figure 1.

2. In Figure 1C-D the cell viability upon SAF treatment is reduced more than using the siRNA1 and the siRNA2 as shown in figure 1E-F (particularly in Hep3B cells), making the results unclear.

Response: Thank you for your professional suggestion. Sorry we have not clearly depicted the results. In the present study, we found MARCH1 play an important role in the development and progression of hepatocellular carcinoma (HCC). However, SAF could target various molecules in regulating the development and progression of  HCC, and MARCH1 is only one of the most targets. In additon, There was a difference effect in the cell viability between the SAF and MARCH1 siRNA because of the differeent concentration. Therefore, the cell viability upon SAF treatment is reduced more than using the siRNA1 and the siRNA2. However, we haved added In all, SAF was partially targeting MARCH1 to inhibit the proliferation in HCC cells in Result 2.1 (page 3, line131 to 132).

3. In Figure 5 the Authors deal with the potential transduction mechanisms involved in the action of SAF, arguing that  PI3K and AKT mediate the downregulation of MARCH1 by SAF. The Authors only showed the regulation of these pathways by SAF but not their connection. A similar comment regards the downregulation of the antiapoptotic Mcl-1/Bcl-2.

Response: Thank you for your constructive and professional comments. In the present study, we found SAF could partially downregulated MARCH1 to inhibit HCC growth; and MARCH1 downregulating further suppressed the activation of PI3K/AKT/β-catenin and Mcl-1/Bcl-2 pathway in vitro and in vivo; and we have verified these results by using MARCH1 siRNA. Moreover, we have found that SAF was not regulating MARCH1 via gene transcription, but via proteasome pathway in the complemented experiments (Figure 1D-1E). However, I agree with your advice, that it is very important to study their connection of the regulation of these pathways by SAF. Therefore, it is worth further exploring for us in the future.

Reviewer 2 Report

This study continues and deepens a previous research in which, the same authors have observed that secalonic 25 acid-F (SAF) suppresses the growth of the hepatoblastoma HepG2, cells. It was fondn that SAF targets MARCH1, that regulates the PI3K/AKT/beta-catenin and the antiapoptotic Mcl-1/Bcl-2 signaling cascades. It was confirmed that SAF inhibits the proliferation and colony formation of HepG2 and Hep3B cells, induces apoptosis, inhibits the cell cycle, the migration and invasion of two cell types. SAF also decreases MARCH1 levels through the downregulation of the PI3K/AKT/β-catenin signaling cascade. Moreover, knockdown of MARCH1 by small interfering. These findings were confirmed in an HCC nude mice model. In this model, the apparent diffusion coefficient  values, derived from magnetic resonance imaging , were increased in tumors after SAF treatment.

This paper confirms and extend previous observations made on HepG2 cells, erroneously referred as HCC cells. They are indeed hepatoblastoma cells. The results are of certain interest, but they present some weaknesses.

1.     The MARCH1 expression has been evaluated by Western blot analysis.  MARCH1, mRNA expression should also be determined.

2.     Quantitative evaluation of immunoblotting analysis of MARCH1, PI3K, p-AKT and beta-catenin, GAPDH normalized, must also be done, the number of experiments must be indicated and statistical analysis must be performed.

3.     To illustrate caspases cleavage, immunoblots showing the bands of both un-cleaved and cleaved caspases must be included in the figure.

4.     A minor point. The indication “beta-cantenin” in the fig. 5 must be corrected.

Author Response

Comments from Reviewer 2:

1. The MARCH1 expression has been evaluated by Western blot analysis.  MARCH1, mRNA expression should also be determined.

Response: Thank you for your constructive suggestion. We have detected MARCH1 mRNA level response on SAF in HepG2 cells by using qRT-PCR in the complemented experiments. However, we found there is a slight upregulation in gene transcription (Figure 1D). Therefore, we speculated that the effect of SAF on MARCH1 downregulation may via degradation of protein. As Maurice Jabbour had reported that MARCH1 expression is partially regulated at a posttranscriptional level by proteasomes and Lysosomal proteases [Jabbour M, Campbell EM, Fares H, Lybarger L. Discrete domains of MARCH1 mediate its localization, functional interactions, and posttranscriptional control of expression. J Immunol. 2009;183(10):6500-12]. And we found MG-132 as a proteasome inhibitor could stabilize MARCH1 in HepG2 cells treated with SAF in the complemented experiments (Figure 1E). Therefore, we agree with your constructive suggestion and it is need us to further explore in the future study. And we have added some relevant supplementary contents in Results 2.1 (page 3, line 118 to 123), in Discussion (page 13, line 345 to 347), and in Materials and Methods 4.11 (page 17, line 533 to 542).

2. Quantitative evaluation of immunoblotting analysis of MARCH1, PI3K, p-AKT and beta-catenin, GAPDH normalized, must also be done, the number of experiments must be indicated and statistical analysis must be performed.

Response: Thanks for the professional and constructive suggestion. We have evaluated quantitation of immunoblotting analysis of MARCH1, PI3K, p-AKT and beta-catenin, and performed statistical analysis, and GAPDH have been normalized in Figures 1-6 of revision. And we have indicated the number of experiments in Materials and Methods and figure legends which are highlighted in red.

3. To illustrate caspases cleavage, immunoblots showing the bands of both un-cleaved and cleaved caspases must be included in the figure.

Response: Thank you for your constructive suggestion.In the present study, sorry we have used the antibodies of cleaved caspase-3 (CST, 9661) and cleaved caspase-7 (CST, 8438). Therefore, immunoblots showing the bands of cleaved caspases, but not both un-cleaved and cleaved caspases. And sorry it might be hard for us to accomplish the experiment beginning with purchase of antibody in 10 days; However, we agree with your constructive and professional suggestion, and we believe it is very important and helpful for us in the future study.

4.A minor point. The indication “beta-cantenin” in the fig. 5 must be corrected.

Response: Sorry we made a minor point in “beta-cantenin”, and we have corrected it into “beta-catenin” in the fig. 5. 

Reviewer 3 Report

Authors investigated the anti-HCC effects of SAF (secalonic acid-F) via thorough  MARCH1 regulation on the PI3K/AKT/β-catenin pathway,

The manuscript is well-written and all the experiments seemed to be done well.

The following issues should be answered by authors.

Regarding apoptosis, there is a gap between assays by flow-cytometry shown in Fig. 2-B/2-D and DNA indexes assay shown in Fig. 4-A-b-c, especially in HepG2 cells using SAF 1.25 and 2.5 μM. This point should be discussed.

Authors only examined SAF effects using hepatocellular carcinoma cell lines, HepG2 and Hep3B. If authors have the effects of SAF on normal cells, especially hepatic cells or others, please describe it.

Author Response

Comments from Reviewer 3:

1. Regarding apoptosis, there is a gap between assays by flow-cytometry shown in Fig. 2-B/2-D and DNA indexes assay shown in Fig. 4-A-b-c, especially in HepG2 cells using SAF 1.25 and 2.5 μM. This point should be discussed.

Response: Thank you for your professional suggestion and we have changed some representative picture. However, in regarding apoptosis analysis, there is a minor difference of apoptosis ratio in HepG2 cells treated with SAF 1.25 and 2.5 μM; and during the process of doing the experiments in flow-cytometry, we may be carelessly clear out the apoptotic cells in washing and centrifuging. Therefore, there is a gap between assays by flow-cytometry shown in Fig. 2-B and DNA indexes assay shown in Fig. 4-A-b-c. However, Thank your for the helpful suggestion, and it is need us to further improving in the future study.

2. Authors only examined SAF effects using hepatocellular carcinoma cell lines, HepG2 and Hep3B. If authors have the effects of SAF on normal cells, especially hepatic cells or others, please describe it.

Response: Thank you for your constructive and professional suggestion. We found there were some effects on normal hepatic cells lines (HL-7702, HHL-5) in response to SAF (data not show); but in vivo, we found SAF could inhibited HCC tumor growth, and there was no side effect in body weight and growth status of nude mice.

Round 2

Reviewer 1 Report

The Authors attempted to address the comments of the reviewer.

Reviewer 2 Report

The authors have satisfactorily met the reviewer ' comments